# Transductive Learning for Textual Few-Shot Classification in API-based Embedding Models

**Pierre Colombo**[*2,6]    **Victor Pellegrain**[*5,6],
**Malik Boudiaf**[1],    **Victor Storchan**[3],    **Myriam Tami**[6],
**Ismail Ben Ayed**[1],    **Celine Hudelot**[6],    **Pablo Piantanida**[4]

[1]ÉTS Montreal, LIVIA, ILLS, Canada   [2]Equall, Paris, France
[3] Mozilla.ai, Paris, France      [4] ILLS, MILA, CNRS, CentraleSupélec, Canada
[5] IRT SystemX Saclay, France, [6] MICS, CentraleSupelec, Universite Paris-Saclay, France

## Abstract

Proprietary and closed APIs are becoming increasingly common to process natural language, and are impacting the practical applications of natural language processing, including few-shot classification. Few-shot classification involves training a model to perform a new classification task with a handful of labeled data. This paper presents three contributions. First, we introduce a scenario where the embedding of a pre-trained model is served through a gated API with compute-cost and data-privacy constraints. Second, we propose a transductive inference, a learning paradigm that has been overlooked by the NLP community. Transductive inference, unlike traditional inductive learning, leverages the statistics of unlabeled data. We also introduce a new parameter-free transductive regularizer based on the Fisher-Rao loss, which can be used on top of the gated API embeddings. This method fully utilizes unlabeled data, does not share any label with the third-party API provider and could serve as a baseline for future research. Third, we propose an improved experimental setting and compile a benchmark of eight datasets involving multi-class classification in four different languages, with up to 151 classes. We evaluate our methods using eight backbone models, along with an episodic evaluation over 1,000 episodes, which demonstrate the superiority of transductive inference over the standard inductive setting.

## 1 Introduction

Recent advances in Natural Language Processing (NLP) have been largely driven by the scaling paradigm (Kaplan et al., 2020; Rosenfeld et al., 2019), where larger models with increased parameters have been shown to achieve state-of-the-art results in various NLP tasks (Touvron et al., 2023; Radford et al., 2019). This approach has led to the development of foundation models such as Chat-GPT (Lehman et al., 2023; Kocoń et al., 2023;

Brown et al., 2020), GPT-4 (OpenAI, 2023), GPT-3 (Brown et al., 2020), T5 (Raffel et al., 2020), and BERT (Devlin et al., 2018), which have achieved unprecedented performance in text classification (Liu et al., 2019b), language modeling, machine translation (Fan et al., 2021), and coding tasks (Chen et al., 2021a).

Despite the success of the scaling paradigm, significant challenges still exist especially when the many practical constraints of real-world scenarios have to be met: labeled data can be severely limited (*i.e.,* few-shot scenario (Song et al., 2022; Ye et al., 2021)), data privacy is critical for many industries and has become the subject of increasingly many regulatory pieces (Commission, 2020, 2016), compute costs need to be optimized (Strubell et al., 2019). Furthermore, these challenges are made even more complex as stronger foundation models are now available only through APIs (*e.g.*, OpenAI's GPT-3, GPT-4 or ChatGPT, Anthropic's Claude or Google's PaLM (Chowdhery et al., 2022)) which has led to some of their parameters being concealed, presenting new challenges for model adaptation (Solaiman, 2023). This paper is centered on the fundamental task of few-shot text classification, specifically focusing on cloud-based/API access. Specifically, we formulate three requirements for API-based few-shot learning (FSL) (see Fig. 1):

**(R1) Black-box scenario.** We focus on learning from models that are opaquely deployed in production to the end-user, who only has access to the end-point of the encoder, *i.e.*, the resulting text embedding produced by the final layer of the network.

**(R2) Low resources / computation time.** AI systems are often required to make rapid predictions at high frequencies in various real-world applications. Therefore, any few-shot classifier used in such scenarios should have a low training and inference time, as well as require minimal computational resources.

---

*These authors contributed equally to this work

**(R3) Limited Data Sharing.** When utilizing API models, data sharing becomes a major concern. In the current landscape, providers are increasingly offering less transparent procedures for training their networks. As a result, users prefer sharing as little information as possible, such as labeling schema and annotated data, to safeguard their data privacy. **Shortcomings of Existing Works.** While numerous previous studies have addressed the popular *few-shot* classification setting, to our knowledge no existing line of work adequately satisfies the three API requirements described above. In particular, prompt-based FSL (Schick and Schütze, 2020a) and parameter-efficient fine-tuning FSL (Houlsby et al., 2019) both require access to the model's gradients, while in-context learning scales poorly with the task's size (*e.g* number of shots, number of classes) (Chen et al., 2021b; Min et al., 2021, 2022; Brown et al., 2020) and requires full data sharing. Instead, we focus on methods that can operate within API-based constraints.

Under **R1**, **R2**, and **R3** requirements, the standard inductive learning (Liu et al., 2022) may be quite limiting. To mitigate the labeled data scarcity while retaining API compliance, we revisit transduction (Vapnik, 1999) in the context of textual few-shot classification. Specifically, in the context of FSL, transductive FSL (Liu et al., 2019a) advocates leveraging unlabeled test samples of a task as an additional source of information on the underlying task's data distribution in order to better define decision boundaries. Such additional source essentially comes for free in many *offline* applications, including sentiment analysis for customer feedback, legal document classification, or text-based medical diagnosis.

Our findings corroborate recent findings in computer vision (Liu et al., 2019a; Ziko et al., 2020; Lichtenstein et al., 2020; Boudiaf et al., 2020; Hu et al., 2021b), that substantial gains can be obtained from using transduction over induction, opening new avenue of research for the NLP community. However, the transductive gain comes at the cost of introducing additional hyperparameters, and carefully tuning them. Motivated by Occam's razor principle, we propose a novel hyperparameter-free transductive regularizer based on Fisher-Rao distances and demonstrate the strongest predictive performances across various benchmarks and models while keeping hyperparameter tuning minimal. We believe that this parameter-free transductive regu-larizer can serve as a baseline for future research.

## Contributions

In this paper, we make several contributions to the field of textual FSL. Precisely, our contributions are threefold:

**A new textual few-shot scenario:** We present a new scenario for FSL using textual API-based models that accurately capture real-world constraints. Our new scenario opens up new research avenues and opportunities to address the challenges associated with FSL using API-based models, paving the way for improved performance in practical applications.

**A novel transductive baseline.** Our paper proposes a transductive FSL algorithm that utilizes a novel parameter-free Fisher-Rao-based loss. By leveraging only the network's embedding **(R1)**, our approach enables fast and efficient predictions **(R2)** without the need to share the labeling schema or the labels of few-shot examples making it compliant with **(R3)**. This innovative method marks a significant step forward in the field of FSL.

**A truly improved experimental setting.** Previous studies on textual few-shot classification (Schick and Schütze, 2022, 2020b; Mahabadi et al., 2022; Tam et al., 2021; Gao et al., 2020) have predominantly assessed their algorithms on classification tasks with a restricted number of labels (typically less than five). We take a step forward and create a benchmark that is more representative of real-world scenarios. Our benchmark relies on a total of eight datasets, covering multiclass classification tasks with up to 151 classes, across four different languages. Moreover, we further enhanced the evaluation process by not only considering 10 classifiers trained with 10 different seeds (Logan IV et al., 2021; Mahabadi et al., 2022), but also by relying on episodic evaluation on 1,000 episodes (Hospedales et al., 2021). Our results clearly demonstrate the superiority of transductive methods.

## 2 Related Work

### 2.1 Few-shot learning in NLP

Numerous studies have tackled the task of FSL in Natural Language Processing (NLP) by utilizing pre-trained language models (Devlin et al., 2018; Liu et al., 2019b; Radford et al., 2019; Yang et al., 2019). These methods can be classified into three major categories: prompt-based, parameter-efficient tuning, and in-context learning.

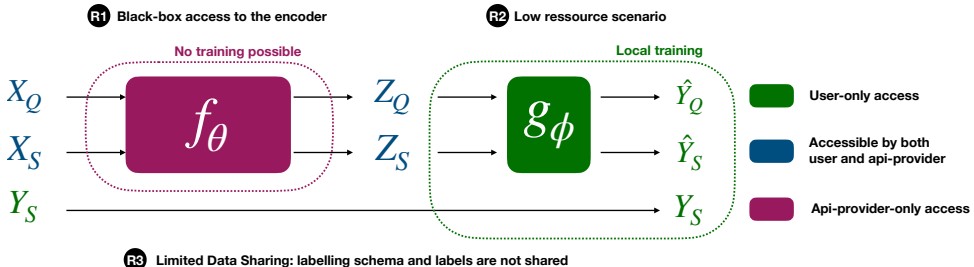

Figure 1: API-based FSL scenario. The black-box API provides embeddings from the pretrained encoder $f_\theta$. The black-box scenario discards existing inductive approaches and in-context learning methods due to the inaccessible of the model's parameters (**(R1)**) and privacy concerns (**(R3)**). This scenario, allows tuning a classification head $g_\phi$ (using induction or transduction) at low computational cost (**R2**) while retaining all support labels locally.

**Prompt-based FSL**: Prompt-based FSL involves the use of natural language prompts or templates to guide the model to perform a specific task (Ding et al., 2021; Liu et al., 2023). For example, the seminal work (Schick and Schütze, 2020a) proposed a model called PET, which uses a pre-defined set of prompts to perform various NLP tasks as text classification. They also impose a choice of a verbalizer which highly impacts the classification performances (Cui et al., 2022; Hu et al., 2021a). However, recent studies have questioned the benefits of prompt-based learning due to the high variability in performance caused by the choice of prompt (Liu et al., 2022). To address this issue, researchers have proposed prompt tuning which involves a few learnable parameters in addition to the prompt (Lester et al., 2021). Nevertheless, these approaches face limitations when learning from API: (i) encoder access for gradient computation is infeasible (as in **R1**), (ii) prompting requires to send data and label which raises privacy concerns (as in **R3**), and (iii) labeling new points is time-consuming (see in **R3**) and expensive due to the need to send all shots for each input token[1].

**Parameter-efficient fine-tuning.** These methods, such as adapters (Houlsby et al., 2019; Pfeiffer et al., 2020), keep most of the model's parameters fixed during training and only update small feed-forward networks that are inserted within the larger model architecture. A recent example is T-FEW (Liu et al., 2022), which adds learned vectors that rescale the network's internal activations. Additionally, it requires a set of manually created prompts for each dataset making it hard to use in practice. Relying on parameter-efficient fine-tuning methods with an API is not possible due to the need to com-

pute gradients of the encoder (as per **R1**) and the requirement to send both the labeling schema and the labels, which violates **R3**.

**In Context Learning (ICL).** In-context learning models are models that utilize input-to-output training examples as prompts to make predictions, without any parameter updates (Wei et al., 2022). These models, such as text-davinci, rely solely on the provided examples to generate predictions, without any additional training. However, a significant drawback of this approach is that the user must supply the input, label examples, and task description, which becomes prohibitively expensive when the number of classes or shots increases, is slow (Liu et al., 2022) (**R2**) and raises data privacy concerns (as highlighted in **R3**). Additionally, the inability to reuse text embeddings for new tasks or with new labels without querying the model's API limits practicality and scalability, making reusable encoding unfeasible for in-context learning models[2].

**Meta-learning.** Meta-learning approaches have for quite long stood as the *de-facto* paradigm for FSL ((Snell et al., 2017; Rusu et al., 2019; Sung et al., 2018b; Lee et al., 2019; Raghu et al., 2019; Sun et al., 2019a)). In meta-learning, the objective is to provide the model with the intrinsic ability to learn in a data-efficient manner. For instance, MAML ((Finn et al., 2017a; Antoniou et al., 2018)), arguably the most popular meta-learning method, tries to train a model such that it can be fine-tuned end-to-end using only a few supervised samples while retaining high generalization ability. Unlike the three previous lines of work, meta-learning methods operate by modifying the pre-training pro-

---

[1]The cost of API queries is determined by the number of input tokens that are transmitted.

[2]Furthermore, as the number of considered classes increases, the fixed size of the transformer limits the number of possible shots that can be fed to the model. Previous studies have often neglected this limitation by focusing on a few numbers of labels.

cedure and therefore assume access to both the training data and the model, which wholly breaks both **R1** and **R3**.

## 2.2 Inductive vs transductive learning

Learning an inductive classifier on embeddings generated by an API-based model, as proposed by (Snell et al., 2017), is a common baseline for performing FSL. This approach is prevalent in NLP, where a parametric model is trained on data to infer general rules that are applied to label new, unseen data (known as inductive learning (Vapnik, 1999)). However, in FSL scenarios with limited labeled data, this approach can be highly ambiguous and lead to poor generalization.

Transduction offers an attractive alternative to inductive learning (Sain, 1996). Unlike inductive learning, which infers general rules from training data, transduction involves finding rules that work specifically for the unlabeled test data. By utilizing more data, such as unlabeled test instances, and aiming for a more localized rule rather than a general one, transductive learning has shown promise and practical benefits in computer vision (Boudiaf et al., 2020, 2021; Ziko et al., 2020). Transductive methods yield substantially better performance than their inductive counterparts by leveraging the statistics of the query set (Dhillon et al., 2019). However, this approach has not yet been explored in the context of textual data.

## 3 API-based Few-shot Learning

### 3.1 Problem Statement

Let $\Omega$ be the considered vocabulary, we denote $\Omega^*$ its Kleene closure. The Kleene closure corresponds to sequences of arbitrary size written with tokens in $\Omega$, *i.e.*, $\Omega^* = \bigcup_{i=0}^{\infty} \Omega^i$. Given an input space $\mathcal{X}$ with $\mathcal{X} \subseteq \Omega^*$ and a latent space $\mathcal{Z}$, we consider a pre-trained backbone model $f_\theta : \mathcal{X} \to \mathcal{Z} = \mathcal{R}^d$, where $\theta \in \Theta$ represents the parameters of the encoder and $d$ is the embedding dimension size. In the API-based setting, we assume that we are unable to access the exact structure of $f_\theta$ as mentioned in **R1**. However, we do have access to the last encoder embedding which is available for our use (see **R1**).

The objective of few-shot classification is to learn a classifier from limited labeled data and generalize it to new, unseen tasks or classes. To accomplish this, randomly sampled few-shot tasks are created from a test dataset $\mathcal{D}_{test} := \{(x_i, y_i)\}_{i=1}^{N_{test}}$

that has a set of unseen classes $\mathcal{Y}_{test}$. Each task involves a few labeled examples from $K$ different classes chosen at random among $\mathcal{Y}_{test}$. These labeled examples constitute the support set $S = \{x_i, y_i\}_{i \in \mathcal{I}_S}$, with a size of $|S| = N_S \times K$. Additionally, each task has an unlabeled query set $Q = \{x_i\}_{i \in \mathcal{I}_Q}$ composed of $|Q| = N_Q \times K$ unseen examples from each of the $K$ classes. $\mathcal{I}_S$ and $\mathcal{I}_Q$ represent the drawn indices during the sampling process for support set and query set, respectively. Pre-trained models use few-shot techniques and the labeled support sets to adapt to the tasks at hand and are evaluated based on their performances on the unlabeled query sets.

*Remark* Setting the values of $N$ and $K$ in textual FSL is not standardized, as discussed in Sec. 3.1. Therefore, in all of our experiments, we have relied on setting $(N, K) \in \{5, 10\}^2$.

### 3.2 Proposed Transductive Method

NLP few-shot classifiers rely only on inductive inference, while computer vision has shown significant performance improvements using transductive inference for FSL. Transductive inference succeeds in FSL because it jointly classifies all unlabeled query samples of a single task, leading to more efficient and accurate classification compared to inductive methods that classify one sample at a time. Let us begin by introducing some basic notation and definitions before introducing our new transductive loss based on the Fisher-Rao distance.

In the API-based few-shot classification setting, our goal is to train a classification head $g_\phi : \mathcal{Z} \to \mathbb{R}^K$ that maps the feature representations to the posterior distribution space for making predictions. To simplify the equations for the rest of the paper, we use the following notations for the posterior predictions of each $i \in \mathcal{I}_S \cup \mathcal{I}_Q$ and for the class marginals within $Q$: $p_{ik} = g_\phi(f_\theta(x_i))_k = \mathbb{P}(Y = k | X = x_i; \theta, \phi)$ and $\widehat{p}_k = \frac{1}{|Q|} \sum_{i \in \mathcal{I}_Q} p_{ik} = \mathbb{P}(Y_Q = k; \theta, \phi)$ where $X$ and $Y$ are the r.v.s associated with the raw features and labels, respectively, and where $Y_Q$ means restriction of the r.v. $Y$ to set $Q$.

For training the classification head in the transductive setting, prior research aims at finding $\phi$ such that $\phi = \arg\min \text{CE} - \lambda \times R_Q$[3], with $\text{CE} := -\frac{1}{|S|} \sum_{i \in \mathcal{I}_S} \sum_{k=1}^{K} y_{ik} \log(p_{ik})$ being the cross-entropy supervision on the support set (in which $y_{ik}$ is the $k^{\text{th}}$ coordinate of the one-hot en-

---

[3] $\lambda$ is set to 1 in all the experiments.

coded label vector associated to sample $i$) and $R_Q$ being a transductive loss on the query set $Q$.

Note that this transductive regularization has been proposed in the literature based on the Info-Max principle (Cardoso, 1997; Linsker, 1988), and the inductive loss can be found by setting $\lambda = 0$. In what follows, we review the regularizers introduced in previous work.

**Entropic Minimization (H)** An effective regularizer for transductive FSL can be derived from the field of semi-supervised learning, drawing inspiration from the approach introduced in (Grandvalet and Bengio, 2004). This regularizer, proposed in (Dhillon et al., 2019), utilizes the conditional Shannon Entropy (Cover, 1999) of forecast results from query samples during testing to enhance model generalization. Formally:

$$R_Q^H = \frac{1}{|Q|} \sum_{i \in \mathcal{I}_Q} \sum_{k=1}^{K} p_{ik} \log(p_{ik}). \qquad (1)$$

**Mutual Information Maximization (I)** A promising alternative to entropic minimization for addressing the challenges of transductive FSL is to adopt the Info-max principle. (Boudiaf et al., 2020) extended this idea, introduced in (Hu et al., 2017), and propose as regularizer a surrogate of the mutual-information $R_Q^I(\alpha) =:$

$$-\sum_{k=1}^{K} \hat{p}_k \log \hat{p}_k + \alpha \frac{1}{|Q|} \sum_{i \in \mathcal{I}_Q} \sum_{k=1}^{K} p_{ik} \log(p_{ik}). \quad (2)$$

**Limitation of existing strategies**: Despite its effectiveness, the previous method has a few limitations that should be taken into account. One of these limitations is the need to fine-tune the weight of different entropies using the hyperparameter $\alpha$. This parameter-tuning process can be time-consuming and may require extensive experimentation to achieve optimal results. Additionally, recent studies have shown that relying solely on the first Entropic term, which corresponds to the Entropic minimization scenario in Equation 1, can lead to suboptimal performance in FSL.

### 3.3 A Fisher-Rao Based Regularizer

In the FSL scenario, minimizing parameter tuning is crucial. Motivated by this, in this section, we introduce a new parameter-free transductive regularizer that fits into the InfoMax framework. Additionally, our loss inherits the attractive properties of

the Fisher-Rao distance between soft-predictions $\mathbf{q} := (q_1, \ldots, q_K)$ and $\mathbf{p} := (p_1, \ldots, p_K)$, which is given by (Picot et al., 2023):

$$d_{\text{FR}}(\mathbf{q}, \mathbf{p}) := 2 \arccos \left( \sum_{k=1}^{K} \sqrt{q_k \times p_k} \right). \quad (3)$$

The proposed transductive regularizer denoted by $R_Q^{\text{FR}}$, for each single few-shot task, can be described as measuring the Fisher-Rao distance between pairs of query samples:

$$R_Q^{\text{FR}} := \frac{1}{|Q|} \sum_{i \in \mathcal{I}_Q} - \log \sum_{j \in \mathcal{I}_Q} \sum_{k=1}^{K} \sqrt{p_{ik} \times p_{jk}} \quad (4)$$

$$= \frac{1}{|Q|} \sum_{i \in \mathcal{I}_Q} - \log \sum_{j \in \mathcal{I}_Q} \cos \left( \frac{d_{\text{FR}}(\mathbf{p}_i, \mathbf{p}_j)}{2} \right), \quad (5)$$

where $d_{\text{FR}}(\mathbf{p}_i, \mathbf{p}_j)$ is the Fisher-Rao distance between pairs of soft-predictions $(\mathbf{p}_i, \mathbf{p}_j)$. Furthermore, it is shown that expression (4) yields a surrogate of the Mutual Information as shown by the following proposition. This result to the best of our knowledge is new, as far as we can tell.

**Theorem 1.** *(Fisher-Rao as a surrogate to maximize Mutual Information) Let* $(\mathbf{q}_i)_{i \in \mathcal{I}_Q}$ *be a collection of soft predictions corresponding to the query samples. Then, it holds that* $\forall\, 0 \leq \alpha \leq 1$:

$$R_Q^{FR} + \log |Q| \leq R_Q^I(1) \leq R_Q^I(\alpha), \quad (6)$$

*Proof:* Further details are relegated to Ap. A.

*Advantage of* $R_Q^{FR}$ *over* $R_Q^I(\alpha)$: Similarly to $R_Q^I(\alpha)$, $R_Q^{\text{FR}}$ can be exploited to maximize the Mutual Information. However, $R_Q^{\text{FR}}$ is parameter-free and thus, it does not require tuning $\alpha$.

### 3.4 Additional Few-shot Inductive Baseline

In addition to the transductive methods of Sec. 3.2, we will explore three additional inductive methods for few-shot classification: prototypical networks, linear probing, and a semi-supervised classifier.

**Prototypical Networks (PT)** PT learn a metric space where the distance between two points corresponds to their degree of similarity. During inference, the distance between the query example and each class prototype is computed, and the predicted label is the class with the closest prototype. PT has been widely used in NLP and is considered as a strong baseline (Snell et al., 2017; Sun et al., 2019b; Gao et al., 2019).

Table 1: Datasets Statistics.

| Dataset | Classes (K) |
|---------|-------------|
| Tweet | 20 |
| Emotion | 25 |
| Amazon | 30 |
| B77 | 77 |
| Clinc | 151 |

**Linear Probing (CE)** Fine-tuning a linear head on top of a pretrained model is a popular approach to learn a classifier for classification tasks and was originally proposed in (Devlin et al., 2018).

**Semi-supervised Baselines (SSL)**. We additionally propose two semi-supervised baselines following two steps. In the first step, a classifier is trained using the support set $\mathcal{S}$ and used to label $\mathcal{Q}$. In the second step, the final classifier is trained on both $\mathcal{S}$ and $\mathcal{Q}$ with the pseudo label obtained from the first step.

## 4 An Enhanced Experimental Setting

### 4.1 Datasets

Benchmarking the performance of FSL methods on diverse sets of datasets is critical to evaluate their generalization capabilities in a robust manner as well as their potential for real-world applications. Previous work on FSL (Karimi Mahabadi et al., 2022; Perez et al., 2021) mainly focuses on datasets with a reduced number of classes (*i.e.*, $K < 5$). Motivated by practical considerations we choose to build a new benchmark composed of datasets with a larger number of classes. Specifically, we choose Go Emotion (Demszky et al., 2020), Tweet Eval (Barbieri et al., 2020), Clinc (Larson et al., 2019), Banking (Casanueva et al., 2020) and the Multilingual Amazon Reviews Corpus (Keung et al., 2020). These datasets cover a wide range of text classification scenarios and are of various difficulty[4]. A summary of the datasets used can be found in Tab. 1.

### 4.2 Model Choice

The selection of an appropriate backbone model is a critical factor in achieving high performance in few-shot NLP tasks. To ensure the validity and robustness of our findings, we have included a diverse range of transformer-based backbone models in our study, including

1. *Three different sizes of RoBERTa based models* (Liu et al., 2019b). Similar to BERT, RoBERTa is

---

[4]Datasets are available in Dataset (Lhoest et al., 2021)

pretrained using the closed task (Taylor, 1953). We consider two different sizes of the RoBERTa model, namely RoBERTa (B) with 124M parameters and RoBERTa (L) with 355M parameters and Distil-RoBERTa, a lighter version of RoBERTa trained through a distillation process (Hinton et al., 2015), for a total of 82M parameters.

2. *Three sentence-transformers encoder* (Reimers and Gurevych, 2019). Following (Muennighoff et al., 2022), we consider MPNET-base (Song et al., 2020), MiniLM (Wang et al., 2020), and Albert Small V2 (Lan et al., 2019).

3. *Multilingual models.* To address realistic multilingual scenarios, we rely on three sizes of XLM-RoBERTa (Conneau et al., 2020, 2019): base (B), large (L) and XL (XL).

4. `text-davinci` *model*: to mimic the typical setting of API-based models, we also conduct experiments on `text-davinci`, only accessible through OpenAI's API.

### 4.3 Evaluation Framework

Prior research in textual FSL typically involves sampling a low number of tasks, typically less than 10, of each dataset. In contrast, we utilize an episodic learning framework that generates a large number of N-shots K-ways tasks. This framework has gained popularity through inductive meta-learning approaches, such as those proposed by (Finn et al., 2017b; Snell et al., 2017; Vinyals et al., 2016; Sung et al., 2018a; Mishra et al., 2017; Rusu et al., 2019; Oreshkin et al., 2018), as it mimics the few-shot environment during evaluation and improves model robustness and generalization. In this context, episodic training implies that a different model is initialized for each generated few-shot task, and all tasks are compiled independently in parallel. This approach allows to the computation of more reliable performance statistics by evaluating the generalization capabilities of each method on a more diverse set of tasks. To account for the model's generalization ability, we average the results for each dataset over 1000 episodes, with the N considered classes varying in every episode. For each experiment, we consider the F1-Score.

## 5 Experiments

### 5.1 Case Study of `text-davinci`

In this experiment, we investigate the performance of `text-davinci` in both its language model and

Table 2: Aggregated performance over K, N, the different datasets for `text-davinci`. $|x|$ stands for the averaged input length.

| N-shots | 10 | | 5 | | $|x|$ |
|---|---|---|---|---|---|
| K-ways | 10 | 5 | 10 | 5 | |
| FR | **69.83** | **77.46** | **66.70** | **75.03** | 14.2 |
| H | 10.00 | 20.00 | 10.01 | 20.04 | 14.2 |
| I | **68.38** | **75.82** | **65.15** | **73.06** | 14.2 |
| CE | 68.21 | 75.47 | 64.92 | 72.70 | 14.2 |
| PT | 67.95 | 75.41 | 64.60 | 72.50 | 14.2 |
| SSL | 68.27 | 75.55 | 64.99 | 72.75 | 14.2 |
| ICL | 68.9 | 76.24 | 65.2 | 74.3 | 900 |

embedding-based model forms. We assess its classification capabilities using the aforementioned baseline and explore the language model's performance when applied in an in-context learning (ICL) setup with prompting.

**Takeaways**. From Tab. 2, we observe that SSL performs comparably to CE, which is simpler to use and will be considered as the baseline in the next part of our study. Although ICL slightly outperforms CE, its implementation comes at a significant cost. In ICL, each class requires N shots, forcing the user to send a long input query with additional instructions. This query length becomes prohibitive as the number of classes increases, and on average, it is 58 times longer than using the embedding base API in our benchmark. The lengthy input and ICL approach make it time-consuming for generation (violating **R1**), require the user to provide labels (violating **R2**), and prevent the reuse of embeddings for future use (*e.g.*, retrieval, clustering). Additionally, ICL is 60 times more expensive than CE. Thus, we will discard ICL for the subsequent part of this study.

## 5.2 Overall Results

**Global results:** To evaluate the effectiveness of various few-shot methods, we conducted a comprehensive analysis of their classification performance across all datasets, all backbones, and all considered N-shots/K-ways scenarios. Results are reported in Tab. 3. *An interesting observation is that transductive approaches I and FR outperform their inductive counterparts (CE and PT).* Notably, we found that vanilla entropy minimization, which solely relies on H, consistently underperforms in all considered scenarios. Our analysis revealed that FR surpasses traditional fine-tuning based on cross-entropy by a margin of 3.7%.

**Mono-lingual experiment**: In order to thoroughly analyze the performance of each method, we conducted a per-dataset study, beginning with a

Table 3: Aggregated performance over K,N, the different datasets and considered backbone.

| N-shots | 10 | | 5 | |
|---|---|---|---|---|
| K-ways | 10 | 5 | 10 | 5 |
| FR | **52.09** | **61.99** | **48.71** | **56.55** |
| I | 50.07 | 59.17 | 46.42 | 55.74 |
| H | 15.07 | 27.39 | 15.33 | 25.84 |
| CE | 48.31 | 56.87 | 45.27 | 53.94 |
| SSL | 50.39 | 58.78 | 47.33 | 55.85 |
| PT | 47.29 | 56.05 | 44.32 | 53.20 |

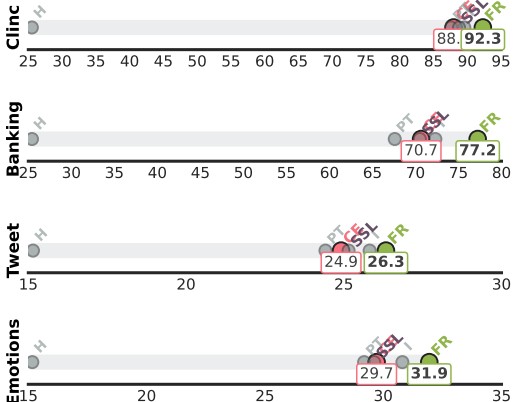

Figure 2: Performance on the monolingual datasets.

focus on the mono-lingual datasets. Fig. 2 reveals that the global trends observed in Tab. 3 remain consistent across datasets of varying difficulty levels. Notably, we observed consistent improvements achieved by transductive regularizers (such as I or FR) over CE. However, the relative improvement is highly dependent on the specific dataset being evaluated. Specifically, FR achieves +6.5% F1-score on Banking, but only a shy +1.5% on Tweet. A strong baseline generally suggests highly discriminative features for the task, and therefore a strong upside in leveraging additional unlabeled features, and vice versa. Therefore, we hypothesize that the potential gains to be obtained through transduction correlate with the baseline's performance.[5]

## 5.3 Study Under Different Data-Regime

In this experiment, we investigated the performance of different loss functions under varying conditions of 'ways' and 'shots'. As shown in Fig. 3, we observed that increasing the number of classes ('ways') led to a decrease in F1 while increasing the number of examples per class ('shots') led to an improvement in F1. This can be explained by

---

[5]Additional multilingual results (*i.e.*, on es, de, fr) can be found on Sec. B.3. They exhibit the same behavior.

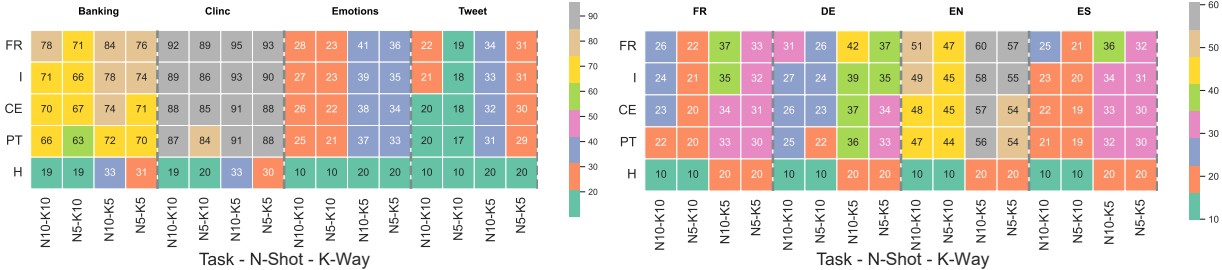

Figure 3: The effect of ways and shots on test performance on monolingual (left) and multilingual (right) datasets.

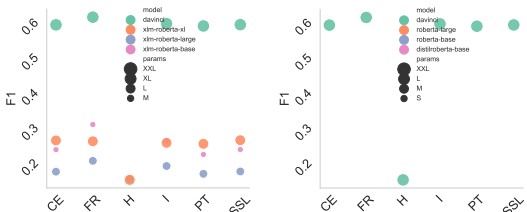

Figure 4: Impact of model size.

the fact that having more data enables the classifier to better discern the characteristics of each class.

Interestingly, the relationship between the number of shots and classification F1 may not be the same for all classes or all loss functions. Fig. 3 shows that different loss functions (e.g. FR on banking) benefited greatly from adding a few shots, while others did not show as much improvement. However, this variability is dependent on the specific dataset and language being used, as different classes may have different levels of complexity and variability, and some may be inherently easier or harder to classify than others.

### 5.4 Ablation Study On Backbones

In this experiment, we examined how different loss functions perform when increasing the number of parameters in various models. The results, presented in Fig. 4, show the average performance across the experiments and are organized by the loss function. We observed an *inverse scaling law* for both the RoBERTa and XLM-RoBERTa family of models, where increasing the number of parameters led to a decrease in performance for the losses tested. However, within the same family, we observe that the superiority of FR remains consistent. An interesting finding from Fig. 4 is that the transductive regularization technique using FR outperforms other methods on text-davinci. This highlights the effectiveness of FR in improving the performance of the model and suggests that transductive regularization may be a promising approach for optimizing language models.

Table 4: Training time for 1 episode on a M1-CPU.

| Loss | CPU Time |
| --- | --- |
| CE | 0.45s |
| FR | 0.83s |
| H | 0.75s |
| I | 0.83s |
| PT | 0.01s |
| SSL | 0.80s |

### 5.5 Practical Considerations

In this experiment, we adopt a practical standpoint and aim to evaluate the effectiveness of an API model, specifically text-davinci. In Tab. 4, we report the training speed of one episode on a MAC with CPU. Overall, we observed that the transductive loss is slower as it necessitates the computation of the loss on the query set, whereas PT is faster as it does not involve any optimization. Furthermore, we note that FR is comparable in speed to I. To provide a better understanding of these results, we can compare our method with existing approaches (in the light of **R2**). For instance, PET (Schick and Schütze, 2020a) entails a training time of 20 minutes on A100, while ADAPET (Tam et al., 2021) necessitates 10 minutes on the same hardware.

### 6 Conclusions

This paper presents a novel FSL framework that utilizes API models while meeting critical constraints of real-world applications (i.e., **R1**, **R2**, **R3**). This approach is particularly appealing as it shifts the computational requirements (**R2**), eliminating the need for heavy computations for the user and reducing the cost of embedding. To provide a better understanding, embedding over 400k sequences cost as low as 7 dollars. In this scenario, our research highlights the potential of transductive losses, which have previously been disregarded by the NLP community. A candidate loss is the Fisher-Rao distance which is parameter-free and could serve as a simple baseline in the future.

# 7 Limitations

We are optimistic that our research will have a positive impact on society. Nonetheless, it is essential to acknowledge the limitations of API-based few-shot classification models despite their promising results in various tasks. Firstly, the performance of the introduced methods is heavily dependent on the quality of available API models. If the API models do not provide sufficient information or lack diversity, the introduced methods may struggle to accurately classify input texts. Secondly, the black-box nature of the backbone limits the interpretability of API-based few-shot classification methods, which may hinder their adoption. Ultimately, the aim of this work is to establish a baseline for future research on transductive inference. As a result, not all existing transductive methods are compared in this study.

## Acknowledgements

This work was performed using HPC resources from GENCI-IDRIS (Grants 2022- AD01101838, 2023-103256 and 2023-101838).

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

# A   Proof of Proposition 1

In this Appendix, we prove the inequality (Eq. 6) provided in Proposition 1. The right-hand side of (Eq. 6) follows straightforwardly from the definition of $R_Q^I(\alpha)$ and the non-negativity of the Shannon entropy. In order to prove the first inequality, we need to introduce the following intermediate result.

For any arbitrary random variable (r.v) $X$ and countable r.v $Y$, and any real number $\beta$, let

$$I_\beta(X;Y) := -\mathbb{E}_{X^\star Y} \log \mathbb{E}_X \left[ \frac{P(Y|X)}{P(Y|X^\star)} \right]^\beta,$$

where the r.v $X^\star$ follows the same distribution than $X$. Notice that it is obvious that $I_1(X;Y) = I(X;Y)$, where $I(X;Y)$ is Shannon Mutual Information.

**Lemma 1.** *For any arbitrary r.v. $X$ and countable r.v. $Y$, we have*

$$I(X;Y) \geq I_\beta(X;Y), \ \ for\ 0 \leq \beta \leq 1.$$

*Proof of the lemma:* We must show that the different of $I(X;Y) - I_\beta(X;Y)$ is nonnegative. To this end, we write this difference as:

$$I(X;Y) - I_\beta(X;Y) \tag{7}$$

$$= -\mathbb{E}_{X^\star Y} \log \frac{P^{1-\beta}(Y|X^\star)\mathbb{E}_X P(Y|X)}{\mathbb{E}_X P^\beta(Y|X)} \tag{8}$$

$$\geq -\log \mathbb{E}_{X^\star Y} \frac{P^{1-\beta}(Y|X^\star)\mathbb{E}_X P(Y|X)}{\mathbb{E}_X P^\beta(Y|X)} \tag{9}$$

$$= -\log \sum_{y\in\mathcal{Y}} \mathbb{E}_{X^\star} P(y|X^\star) \frac{P^{1-\beta}(y|X^\star)\mathbb{E}_X P(y|X)}{\mathbb{E}_X P^\beta(y|X)} \tag{10}$$

$$= -\log \sum_{y\in\mathcal{Y}} \frac{\mathbb{E}_{X^\star} P^\beta(y|X^\star)\mathbb{E}_X P(y|X)}{\mathbb{E}_X P^\beta(y|X)} \tag{11}$$

$$= -\log \sum_{y\in\mathcal{Y}} \mathbb{E}_X P(y|X) \tag{12}$$

$$= 0, \tag{13}$$

where the first inequality follows by applying Jensen's inequality to the function $t \mapsto -\log(t)$.

*Proof of Proposition 1:* From Lemma 1, using Jensen's inequality, we have

$$I(X;Y) = -\mathbb{E}_{X^\star Y} \log \mathbb{E}_X \left[ \frac{P(Y|X)}{P(Y|X^\star)} \right], \tag{14}$$

$$\geq -\mathbb{E}_{X^\star Y} \log \mathbb{E}_X \left[ \frac{P(Y|X)}{P(Y|X^\star)} \right]^\beta \tag{15}$$

$$\geq -\mathbb{E}_{X^\star} \log \mathbb{E}_X \mathbb{E}_{Y|X^\star} \left[ \frac{P(Y|X)}{P(Y|X^\star)} \right]^\beta \tag{16}$$

$$= -\mathbb{E}_{X^\star} \log \mathbb{E}_X \tag{17}$$

$$\sum_{y\in\mathcal{Y}} P^\beta(Y|X) P^{1-\beta}(Y|X^\star), \tag{18}$$

where inequality (15) follows by applying Lemma 1 and inequality (16) follows by exploiting the convexity of the function $t \mapsto -\log(t)$ for any $0 \leq \beta \leq 1$. Finally, it is not difficult to check from the definition of the Fisher-Rao distance given by expression (3) that

$$\cos\left( \frac{d_{\mathrm{FR}}(P(y|X=x), P(y|X=x^\star))}{2} \right) = \tag{19}$$

$$\sum_{y\in\mathcal{Y}} \sqrt{P(y|X=x)P(y|X=x^\star)}. \tag{20}$$

Table 5: Preliminary experiment results. Accuracy of the different backbone.

| Model | Params | Emotion | Twitter | Clinic | Banking | | Amazon | |
|---|---|---|---|---|---|---|---|---|
| | | en | en | en | en | en | fr | es | de |
| Albert Small V2 (XS) | 11M | 25.2 | 18.3 | 67.0 | 88.1 | 33.5 | X | X | X |
| MiniLM (S) | 33M | 30.2 | 19.3 | 67.1 | 92.3 | 39.5 | X | X | X |
| MPNET-base (B) | 109M | 30.2 | 22.5 | 67.4 | 94.3 | 41.3 | X | X | X |
| DistilRoBERTa (S) | 82M | 23.3 | 26.0 | 68.5 | 90.9 | 40.0 | X | X | X |
| RoBERTa (B) | 124M | 21.0 | 25.5 | 66.7 | 91.4 | 39.2 | X | X | X |
| RoBERTa (L) | 355M | 15.0 | 23.0 | 64.5 | 90.0 | 38.1 | X | X | X |
| XLM-RoBERTa (B) | 278M | 21.0 | 22.1 | 66.5 | 87.0 | 40.1 | 19.2 | 17.5 | 18.3 |
| XLM-RoBERTa (L) | 559M | 14.0 | 18.0 | 64.5 | 86.2 | 38.2 | 17.5 | 15.6 | 18.1 |
| XLM-RoBERTa (XL) | 3.48B | 25.4 | 19.0 | 68.9 | 95.0 | 41.0 | 18.9 | 17.9 | 22.0 |
| text-davinci | 175B | 38.9 | 35.3 | 70.4 | 98.7 | 48.4 | 30.4 | 34.0 | 33.5 |

Using the identity given by (19) in expression (18), and setting $\beta = 1/2$, we obtain the following lower bound on $I(X;Y)$:

$$-\mathbb{E}_{X^\star} \log \mathbb{E}_X \cos\left(\frac{d_{\mathrm{FR}}(P(y|X), P(y|X^\star))}{2}\right)$$

The inequality (6) immediately follows by replacing the distribution of the r.v. $X$ with the empirical distribution on the query and $P(y|x)$ with the soft-prediction corresponding to the feature $x$, which concludes the proof of the proposition.

# B  Additional Experimental Results

## B.1  Preliminary Classification Results

**Preliminary Experiment.** In our experiments, the backbone models are of utmost importance. Our objective in this preliminary experiment is to assess the efficacy of these models when fine-tuning **only** the model head across a variety of datasets. Through this evaluation, we aim to gain insight into their generalization abilities and any dataset-specific factors that may influence their performance. This information can be utilized to analyze the performance of different models in the few-shot scenario, as described in Sec. 5. We present the results of this experiment in Tab. 5, noting that all classes were considered, which differs from the episodic training approach detailed in Sec. 5.

## B.2  A Dive Into text-davinci results

text-davinci appears to be the backbone providing the most informative a priori embeddings in Tab. 5 and could be considered as the prime model for API-based FSL, showcasing the current requirements in this area. It is thus a typical candidate for application uses that must meet the following criteria **(R1)** - **(R3)**. Therefore, we put a special emphasis on its related results.

Fig. 6 (top) details the text-davinci results of the experiments conducted on the mono-lingual datasets. These plots highlight the consistency of the tendencies that emerged in Tab. 5, Tab. 3 and Fig. 2, namely: the superiority of transductive approaches ($FR$ and $I$) over inductive ones ($CE$ and $PT$), the underperformance of the entropic-minimization-based strategy (H), and the higher amount of information conveyed by text-davinci learned embeddings over other backbones, resulting in higher F1 scores on all datasets.

These phenomena still occur in the multi-lingual setting, as illustrated in Fig. 6 (bottom), stressing the superiority of transductive (and especially FR) over other approaches for presumably universal tasks, beyond English-centered ones, and without the need for using language-specific engineering as for prompting-based strategies.

Note that for both of these settings, the entropic-minimization-based strategy (H) seems to be capped at a 15% F1 score, thus with no improvement over other backbones embeddings, and independently of the dataset difficulty.

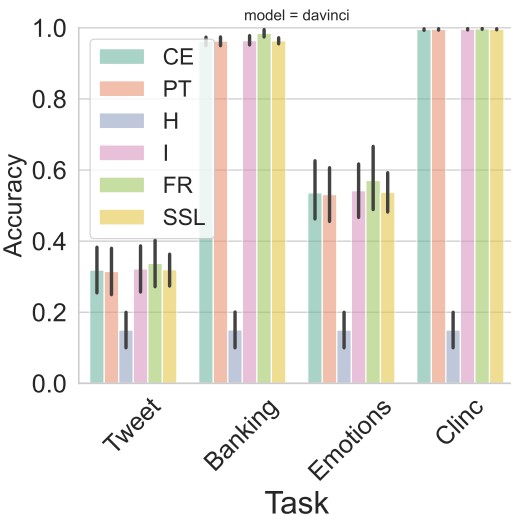

Figure 6: The different losses when training a on `text-davinci` embeddings.

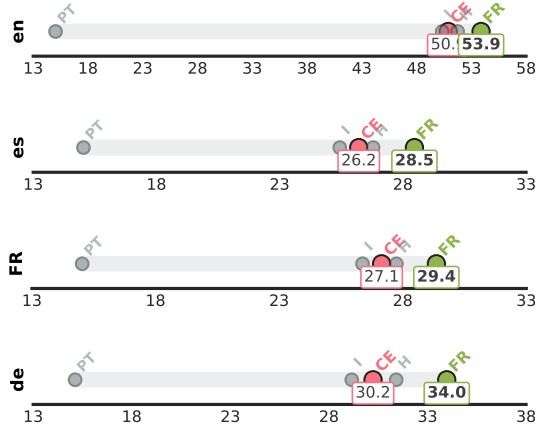

Figure 8: Losses Performance on multilingual datasets.

## B.3 Multilingual Experiment

To provide an exhaustive analysis, we report the same experiment that is made in Sec. 5.2, for multi-lingual models on the Amazon dataset. While both Latin languages (French and Spanish) share close results, with an F1 gain of 2.8% for FR over CE, the results in the German and English language exhibit an F1 increased by almost 4%.

## B.4 Importance of Model Backbones on Monolingual Experiment

In this section, we report the results of our experiment aggregated per backbone. The goal is to understand how the different losses behave on the different backbones. The results are presented in Fig. 10. While the trends observed in the previous charts are retrieved for the majority of backbones, some of these models are exceptions. For example, while transductive methods perform generally better than inductive methods, the CE-based method seems to perform slightly better than I for XLM-RoBERTa-xl. Additionally, while FR is the most effective method for the majority of backbones, it is surpassed by I for the all-distilroberta-v1 model. Furthermore, the inverse-scaling-law details are found for the RoBERTa(B/L) and XLM-RoBERTa (B/L) models per dataset. In general, it is interesting to note that although model performance is constrained by dataset difficulty, the performance order of each method is consistent across all 4 datasets for each considered backbone.

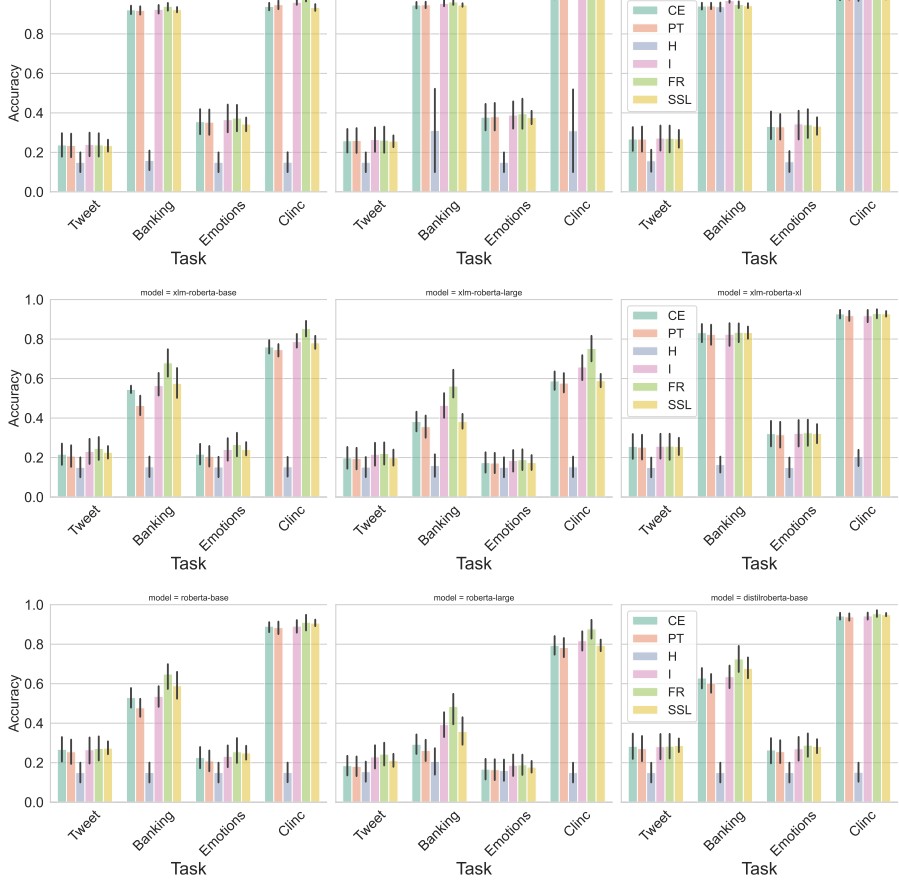

Figure 10: Performance on monolingual datasets.

### B.4.1 Results Per Language

In this experiment, we report the performance of different losses on the Amazon dataset by averaging the results over the number of shots, ways, and model backbones. The results are presented in Tab. 6. Our observations indicate that the transductive regularization improves the results for all languages over the inductive baseline (i.e., CE), with a substantially higher gain for the German language. Additionally, we note that the observed improvements for FR are more consistent. This further demonstrates that the transductive loss can be useful in few-shot NLP. In the future, we would like to explore the application of transductive inference to other NLP tasks such as sequence generation (Pichler et al., 2022; Colombo et al., 2019, 2021d,b) and classification tasks (Chapuis et al., 2020; Colombo et al., 2022d,b; Himmi et al., 2023) as well as NLG evaluation (Colombo et al., 2021e, 2022c, 2021c,a,b) and Safe AI (Colombo et al., 2022a; Picot et al., 2022a,b; Darrin et al., 2022, 2023).

|      | fr        | de        | en        | es        |
|------|-----------|-----------|-----------|-----------|
| FR   | **29.36** | **33.98** | **53.89** | **28.47** |
| I    | 27.74     | 31.41     | 51.75     | 26.79     |
| H    | 15.04     | 15.13     | 15.04     | 15.04     |
| CE   | 27.15     | 30.24     | 50.89     | 26.21     |
| PT   | 26.37     | 29.16     | 50.34     | 25.44     |
| SSL  | 27.20     | 30.29     | 50.94     | 26.26     |

Table 6: Global Results for multilingual Amazon