# OpenReview forum: "Transductive Learning for Textual Few-Shot Classification in API-based Embedding Models"
_EMNLP/2023/Conference — EMNLP 2023 Main_

### Official Review · Reviewer_CwfZ · 2023-08-03

**Soundness:** 3

**Excitement:**

3: Ambivalent: It has merits (e.g., it reports state-of-the-art results, the idea is nice), but there are key weaknesses (e.g., it describes incremental work), and it can significantly benefit from another round of revision. However, I won't object to accepting it if my co-reviewers champion it.

**Paper Topic And Main Contributions:**

The paper propose an API based setting for few-shot learning, where the encoder is concealed in a black-box API and a classifier is trained on the API output embeddings. Further, it considers the transductive setting that assumes the query data being available during training.

**Reasons To Accept:**

The paper is generally written clearly.

The proposed setting is reasonable and the experiment results seem promising.

**Reasons To Reject:**

The technique novelty of the paper concentrates in Section 3.3, i.e., the Fisher-Rao regularizer, which is insufficiently explained. Especially, what is the intuition that we have to compute the Fisher-Rao distance between each pair of the query instance and how it can work as a regularizer? I think these are missing points that are important for readers to trust the experimental results.

In the theorem, both the proposed $R_Q^{FR}$ and $R_Q^{I}(1)$ can lower-bounds $R_Q^{I}(\alpha)$. And, it seems from (6) that the $R_Q^{I}(1)$ is a tighter lower-bound. Then, why we want to use $R_Q^{FR}$ instead of $R_Q^{I}(\alpha)$, which is also non-parametric?

**Reproducibility:**

3: Could reproduce the results with some difficulty. The settings of parameters are underspecified or subjectively determined; the training/evaluation data are not widely available.

**Reviewer Confidence:**

3: Pretty sure, but there's a chance I missed something. Although I have a good feel for this area in general, I did not carefully check the paper's details, e.g., the math, experimental design, or novelty.

---

> ### Author Rebuttal · Authors · 2023-08-28
>
> We express our gratitude to Reviewer CwfZ for their reading of the manuscript. **We are glad that they find the paper clear and find the novel problem and experimental results promising.**
>
> Below we answer the question on the Fisher Rao regularizer:
>
> **The intuition behind using the pairwise Fisher-Rao distance as a  transductive regularizer**
>
> We really appreciate this comment and we totally agree that giving additional intuitive arguments as to why the transductive Fisher-Rao (FR) loss in Eq. (4) works is important for readers. We assumed that Theorem 1, which shows that the FR loss is a proxy for maximizing the mutual information in Eq. (2), is enough. However, we realize that we did not provide intuition as to why maximizing the mutual information in Eq. (2) is relevant (assuming that this is known in computer vision, but we agree with the reviewer that it needs clarification for an NLP audience). In the following, we provide (1) intuitive arguments for maximizing the mutual information in Eq. (2); and (2) direct intuitive arguments, beyond Theroem 1, based on the pairwise distances in the FR loss in Eq. (4), and how it encourages prediction confidence and class balance (similar to maximizing the mutual information). This will also help clarify the advantage of FR over the mutual information (i.e., the adaptive trade-off between prediction confidence and class balance in FR).  We will add these explanations to the paper:
>
> **1- Intuitive arguments for maximizing the mutual information:** Minimizing the conditional entropy (second term in the mutual info in Eq. 2)  encourages the model to output confident predictions close to the vertices of the simplex (i.e. close to one-hot encoding vectors). This conditional-entropy loss is a clustering objective as it models the cluster assumption: The classifier’s boundaries should not occur at dense regions of the unlabeled features . The marginal-entropy loss (the first term in the mutual information in Eq. 2) encourages the distribution of labels to be uniform, thereby avoiding degenerate solutions assigning all samples to one single class.
>
> **2-  Direct intuition behind minimizing the pairwise FR distances in Eq. 4 (without Th1 linking to the mutual info):** From Eq. (4), one could see that maximizing FR minimizes the pairwise dot products between  $\sqrt{{\mathbf p}_i}$ and $\sqrt{{\mathbf p}_j}$ for probability simplex predictions ${\mathbf p}_j$ and ${\mathbf p}_j$. This dot product is the Bhattcharryya coefficient between ${\mathbf p}_i$ and ${\mathbf p}_j$, and it takes values in $[0, 1]$: 1 when ${\mathbf p}_j$ = ${\mathbf p}_j$  and 0 when simplex vectors ${\mathbf p}_j$ and ${\mathbf p}_j$ are orthogonal. Therefore, maximizing FR encourages sparsity of probability predictions ${\mathbf p}_j$ and hence confidence, pushing them towards the vertices of the simplex (similarly to the conditional entropy term $R_Q^H$ in the mutual info in Eq. 2). Intuitively, unconfident predictions in the middle of the simplex (i.e. close to the uniform distribution) lead to high pairwise dot products (close to 1), which decrease the FR loss in Eq. (4) and hence correspond to undesirable solutions. However, unlike conditional entropy $R_Q^H$, the FR loss also penalizes degenerate, extremely imbalanced solutions that assign all points to a single class. Indeed, even with confident predictions, assigning all points to a single class yields high pairwise dot products and, therefore, decreases the FR loss. In summary, the FR loss has it all, both prediction confidence (cluster assumption) and class balance in one single loss, with the trade-off between them being adaptive (more explanation on this below). This is an important advantage over maximizing the mutual information with fixed $\alpha$.
>
>
>
> **Why use $R_Q^FR$ instead of $R_Q^I(1)$, which is a non-parametric, tighter bound?**
>
> We tried to set $\alpha=1$ in our experiment, i.e., using the Mutual Information $R_Q^I(1)$, but it yields suboptimal results, which are lower than those we obtained with the $R_Q^I(0.1)$ loss for example (similar results are reported in Boudiaf et al. 2020). Indeed, fixing the confidence/class balance tradeoff parameter $\lambda$ during gradient-based optimization of the mutual info may yield degenerate solutions (e.g. extremely imbalanced solutions). Assume, for instance, that the current solution is confident but extremely imbalanced. Then, if $\lambda$ is not small enough, the conditional entropy term (second term in the mutual info in Eq. 2) may act as a strong barrier, preventing moving away from such a degenerate solution. On the contrary, in FR in Eq. (4), when the predictions are close to the vertices of the simplex (i.e. confident predictions close to one-hot encoding vectors), one could see that FR in Eq. (4) is approximately the marginal entropy term (first term in the mutual info in Eq. 1), up to a constant. Intuitively, this is akin to adaptively choosing a very small $\lambda$ in the mutual information to penalize the trivial one-class solution. This adaptive confidence/class-balance tradeoff is the main advantage of FR over the mutual information in Eq. (2).
>
> Following the reviewer's comments, we will make sure to include more details in the future version of the manuscript.
>
> **We hope our answers address all concerns of reviewer CwfZ, and we hope they would be keen to consider raising their score.**

---

### Official Review · Reviewer_d1BR · 2023-08-05

**Soundness:** 4

**Excitement:**

4: Strong: This paper deepens the understanding of some phenomenon or lowers the barriers to an existing research direction.

**Paper Topic And Main Contributions:**

This paper formulates a new Few-Shot Learning problem for text classification: In the proposed scenario, one can only access a deployed foundation model through APIs, without seeing the internal states of the model; one submits text sequences to the foundation model and receives embeddings of the text. Then, Few-Shot Learning methods are applied to the text embeddings to perform a text classification task.

The paper discusses several requirements of the considered scenario, and argues that the proposed solution meets the requirements most in the real world. Further, the proposed solution emphasizes a new transductive learning regularizer using the Fisher-Rao distance.

Evaluation is conducted on real-world applications, as well as in controlled studies. The proposed transductive learning regularizer has exhibited its usefulness in the evaluation.

**Reasons To Accept:**

It is good work tackling a timely problem, with solid evaluation conducted in real-world settings. The proposed transductive learning regularizer using the Fisher-Rao distance is new and has good performance.

**Reasons To Reject:**

In the proposed scenario, one has to submit every piece of text to the foundation model in order to obtain the embedding. I don't see why this is related to the Few-Shot Learning setting, because whether there is a label or not, the label is not shared with the foundation model. So if one has a lot of labeled data locally, one can simply perform the normal supervised learning, but using the text embeddings through the foundation model API. Why is the proposed scenario related to Few-Shot Learning?

**Reproducibility:**

4: Could mostly reproduce the results, but there may be some variation because of sample variance or minor variations in their interpretation of the protocol or method.

**Reviewer Confidence:**

4: Quite sure. I tried to check the important points carefully. It's unlikely, though conceivable, that I missed something that should affect my ratings.

---

> ### Author Rebuttal · Authors · 2023-08-28
>
> We express our gratitude to Reviewer d1BR for their meticulous evaluation of the manuscript. **We appreciate their recognition of the timeliness and relevance of the chosen problem for the community. Additionally, we are pleased that they concur with the novelty of our technical contribution regarding the Fisher Rao method.**
>
>
> Below we clarify our positioning:
>
> In our scenario, we work with a limited quantity of locally labeled data (referred as the support set in section 3.1 in the problem statement. The distinguishing feature of our approach lies in its treatment of the target set. Unlike the conventional NLP setting, which predominantly relies on inductive inference, overlooking the distributional information embedded in predictions, our regularizers leverage the distributionnal information within the query set. This allows our method to operate in a transductive manner, making effective use of the (unlabeled) query set.
>
> It is essential to highlight that (as Reviewer d1BR suggests) our method remains flexible enough to adapt to scenarios where a larger pool of labeled data is accessible. However, this particular aspect goes beyond the immediate scope of this paper and is not studied in the paper which rely on N-shots (5 or 10; please see l318) per class. This is particularly relevant when using APIs. If a substantial volume of labeled training data is available, training a model locally might be a more suitable approach. We will add this suggestion in the future research directions of the paper.
>
> **We hope our answers address all concerns of reviewer d1BR, and we hope they would be keen to consider raising their score.**

---

### Official Review · Reviewer_RYzB · 2023-08-11

**Soundness:** 3

**Excitement:**

4: Strong: This paper deepens the understanding of some phenomenon or lowers the barriers to an existing research direction.

**Justification For Ethical Concerns:**

_

**Missing References:**

_

**Paper Topic And Main Contributions:**

The paper solves the few-shot classification problem. Its main contributions are:
1. they defined a model where the embedding of a pre-trained model is calculated as a black-box, via API, without accessing the models internals, and in addition without providing the labels (privacy constraint).
2. they extended the transductive inference paradigm that was previously used at computer vision field to the NLP.
3. in addition to the theoretical proof of the feasibility of their approach the team evaluated the new method by running a wide set of experiments and comparing its performance to broad set of alternative methods, on a multi-language benchmark of eight datasets involving multi-class classification with up to 151 classes.

**Questions For The Authors:**

Why is it enough to show the Theorem 1 only at one direction - R^{FR} <= R^{I}(\alpha}? for example I think we can say that 0 <= R^{I}(\alpha} because of non-negativity of Shanon entropy - but we cant use 0 as regularizer. What do I miss?


**Reasons To Accept:**

All the three paper contributions may have a strong positive impact on the research field and should be shared with the NLP community:
1. The model defined at the paper meets the needs of real life scenarios where strong pre-trained models are executed by third parties and are available only by API and there are strong constraints of sharing private date with that third parties.
2. The ransductive inference paradigm that was previously overlooked by the NLP community looks like a powerful tool and I believe it can be used not only at framework of this paper.
3. the benchmark and the FSL approach presented at this paper can be used to evaluate future works at this field by using the presented FSL approach as a baseline.

**Reasons To Reject:**

I don't see any reason not to accept this paper

**Reproducibility:**

4: Could mostly reproduce the results, but there may be some variation because of sample variance or minor variations in their interpretation of the protocol or method.

**Reviewer Confidence:**

4: Quite sure. I tried to check the important points carefully. It's unlikely, though conceivable, that I missed something that should affect my ratings.

**Typos Grammar Style And Presentation Improvements:**

Page 3: can the colors tones be changed in a way that the figure 1 will be readable at grayscale also? it is hard to distinguish between colors at BW printed version.
Page 4: the usage of footnote at line 346 looks a little bit confusing - since on the first look it looks like a R_{Q} in a third degree.
Page 7: it will be better to use the same order at tables 2 and 3
Page 8: the figure 4 i very hard to read - maybe it worth to move the label to outside of the chart
Page 14: The line 1125 is broken - the formula overlaps the text of the second column
Page 15: to change the order of table 6 to be aligned with 2 and 3
Page 16: at figure 8 the 'FR' should be in lowercase

---

> ### Author Rebuttal · Authors · 2023-08-28
>
> We thank reviewer rYZB for their careful reading of the manuscript. **We are glad they acknowledge that our contributions may have a strong positive impact on the research field and that he does not see any reason not to accept the paper.**
>
> Below we answer their question on the Theorem 1:
>
> As stated by Theorem 1, we have: $R^{FR}_{Q} + \log |Q| \leq R_{Q}(1) = I(X_{Q},Y_{Q}) \leq R_{Q}(\alpha)$
> Thus,  maximizing  $R^{FR}_{Q}$ could be viewed as a proxy for maximizing the mutual information (as we maximize a lower bound on the mutual information).
>
> The confusion of Reviewer rYZB  may arise from the fact that R_{Q}^I(1) = I(X_{Q},Y_{Q}) and not \( R_{Q}^I(1) = -I(X_{Q}, Y_{Q}). To enhance readability, we will add a line of clarification in Eq. 2:
>  \hat{H}(Y_{Q}) - \alpha \hat{H}(Y_{Q} | X_{Q}) = I_{\alpha}(Y_{Q},X_{Q})
> Thus,  -R^{FR} \geq -I_{\alpha}(Y_{Q},X_{Q}) , and by minimizing -R^{FR}, we minimize an upper bound on the mutual information.
>
> **We thank the reviewer for their improvement suggestions that we will make sure to incorporate in order to enhance the paper's quality. We hope our answers address all concerns of reviewer rYZB, and we hope they would be keen to consider raising their score.**

---

### Meta-Review · Area_Chair_y8QQ · 2023-09-19

**Recommendation:** 4

**Metareview:**

This paper discusses the problem of few-shot classification for NLP by using black-box APIs. The setting is very relevant to the current state, where many foundational models are served by companies only through APIs without sharing their internals representations or model architecture details. The authors proposes to use a Transductive Learning approach through a Fisher-Rao regularizer.
The reviewers agree that the paper is timely with respect to the current state of the art. The reviewers agree that the proposed approach is interesting, the experiments are showing the efficacy of the appraoch and that in general  the paper can provide a positive impact to the NLP community.

---

### Decision · Program_Chairs · 2023-10-07

**Decision:**

Accept-Main

**Comment:**

This paper discusses the problem of few-shot classification for NLP by using black-box APIs. The setting is very relevant to the current state, where many foundational models are served by companies only through APIs without sharing their internals representations or model architecture details. The authors proposes to use a Transductive Learning approach through a Fisher-Rao regularizer.
The reviewers agree that the paper is timely with respect to the current state of the art. The reviewers agree that the proposed approach is interesting, the experiments are showing the efficacy of the appraoch and that in general  the paper can provide a positive impact to the NLP community.